# Performance of Imidazoquinoline Glycoconjugate BAIT628 as a TLR7 Agonist Prodrug for Prostate Cancer

**DOI:** 10.3390/ph18060804

**Published:** 2025-05-27

**Authors:** Seyedeh A. Najibi, S. M. Al Muied Pranto, Muhammad Haroon, Amy E. Nielsen, Rock J. Mancini

**Affiliations:** 1Department of Chemistry and Biochemistry, Miami University, 651 E. High Street, Oxford, OH 45056, USA; 2Astante Therapeutics Inc., 201 E. Fifth Street, Cincinnati, OH 45202, USA; amy@astantetherapeutics.com

**Keywords:** prostate cancer, imidazoquinoline, glycoconjugate, Toll-Like Receptor

## Abstract

Despite broad anti-cancer efficacy as Toll-Like Receptor (TLR) 7/8 agonists, imidazoquinolines remain limited in use via systemic administration or in situ vaccination therapies due to inflammatory toxicity. One approach to address this challenge involves better targeting the action of imidazoquinolines by caging them as glycoconjugate prodrugs. Within cancer cells, imidazoquinoline glycoconjugates are activated by hydrolases prior to efflux by ABC transport proteins, where they then elicit tumoricidal effects from the assistance of bystander immune cells, such as tumor-infiltrating lymphocytes and associated macrophages, in local proximity. While this concept of Bystander-Assisted ImmunoTherapy (BAIT) has been established at a molecular level in vitro, tolerability or efficacy of BAIT has not been reported in vivo. Here, we evaluate the MTD and tumor growth delay efficacy of a lead BAIT prodrug (BAIT628) in a male C57BL/6 mouse TRAMP-C2 prostate cancer model to further establish this methodology. Overall, we find that systemic BAIT628 is well tolerated at over 5-fold the dose-limiting inflammatory toxicity of the parent imidazoquinoline (up to 5 mg/mouse/day I.P. for 10 days). Analyzing serum cytokines reveals that IL-10 production, elicited by the mannoside caging group, likely contributes to the enhanced MTD. Using BAIT628 as an in situ vaccination immunotherapy (seven times over 3 weeks) resulted in significant tumor growth delay and increased survival, both alone and in combination with a murinized α-PD-L1 checkpoint blockade. The tumor histology of tumor-infiltrating immune cell subsets (CD4^+^, CD8^+^, CD11c^+^) reveals significant increases in CD11c^+^ populations, consistent with TLR7/8 agonism. Overall, BAIT628 is well tolerated and exhibits significant efficacy in the TRAMP-C2 model. These results demonstrate how the BAIT approach can optimize imidazoquinolines for in vivo tolerability and subsequent efficacy as cancer immunotherapeutics.

## 1. Introduction

Imidazoquinoline Toll-Like Receptor (TLR) 7/8 agonists have long been viewed as promising immunotherapeutic agents [1,2,3]. They elicit broad anti-tumor immunogenicity across a range of cancers [4], act in synergy with other immunotherapies [5], and orthogonally enhance chemotherapeutic efficacy [6]. However, the dose-limiting systemic inflammatory toxicity and profound immune-related adverse events elicited by imidazoquinolines are equally well established, particularly during the translational phase of their development as cancer immunotherapeutics [7].

Attempts to address this challenge commonly involve drug delivery approaches that incorporate active imidazoquinolines into macromolecular architectures, including nanoparticles [8,9], coacervates [10], liposomes [11], polymeric hydrogels [12], and antibody-drug conjugates [13]. Some techniques also cage imidazoquinolines as macromolecular prodrugs (Figure 1A) [14]. While these approaches have been successful in early in vitro and in vivo development stages, the small-molecule imiquimod remains the only successfully FDA-approved imidazoquinoline to date. That said, some comparable scaffolds (such as the imidazopyrimidine series) have more recently been demonstrated to offer computationally predictable immune responses [15], and small-molecule prodrugs of some non-imidazoquinoline TLR 7 agonists (e.g., ruzotolimod) are well tolerated in vivo and are rapidly progressing through clinical trials as well [16]. Together, this suggests that using a more molecularly precise prodrug approach with the imidazoquinolines could likewise help mitigate the systemic inflammatory toxicity that challenges their broader advancement as cancer immunotherapeutics.

Drawing inspiration from the field of chemotherapeutic drug delivery, our group was the first to report a small-molecule imidazoquinoline prodrug [17]. In vitro, we have subsequently demonstrated that imidazoquinoline glycoconjugate prodrugs undergo a unique mechanism of action, termed Bystander-Assisted ImmunoTherapy (BAIT), across melanoma, breast, and prostate cancer cell lines (Figure 1B) [18].

Briefly, BAIT leverages the irregular carbohydrate processing [19,20] and drug efflux [21] inherent to cancer cells. Many therapies inhibit cancer carbohydrate metabolism [22,23], and others target or hijack it [24,25]. However, active drugs liberated in this way are susceptible to drug efflux by ABC transport proteins, particularly P-glycoprotein (MDR1) [26]. Following efflux, these drugs diffuse to interact with cells in local proximity via processes cumulatively termed the bystander effect [27]. Drug efflux lowers intracellular drug concentration, and therefore efficacy, of traditional cytotoxic chemotherapeutics. Conversely, drug efflux enhances the immune response of imidazoquinoline glycoconjugate prodrugs, as the effluxed active drug diffuses to interact with bystander immune cells in local proximity [28]. However, in vivo tolerability or efficacy of the BAIT approach has not been reported. Here, we evaluate the in vivo tolerability and therapeutic efficacy of imidazoquinoline glycoconjugate prodrug BAIT628, both alone and in combination with α-PD-L1 checkpoint inhibitor in a mouse transgenic adenocarcinoma of the mouse prostate (TRAMP) [29] cancer model (Figure 1C).

## 2. Results and Discussion

From our initial proof-of-concept studies [30], we identified a lead BAIT prodrug (BAIT628) that was well metabolized by prostate cancer cells in vitro. The prodrug is a mannoside-imidazoquinoline glycoconjugate that is liberated by α-mannosidases (α-man), including the MAN2C1 isoform, differentially expressed by many prostate cancers [31,32]. Furthermore, following our in vitro study, other groups have subsequently reported the beneficial effects of intra-tumoral mannose [33] as well as the anti-inflammatory result of activating the mannose receptor CD206 in the context of cancer [34]. This indicates that caging an imidazoquinoline as a mannoside prodrug, like BAIT628, could be particularly well tolerated.

As such, our study began with the synthesis of BAIT628, obtained in similar fashion as our previous report. Here, we also optimized the final deprotection step of the synthesis by replacing the sodium methoxide used in the modified Zemplén conditions (which competitively degraded BAIT628’s carbamate) with less nucleophilic KCN (Scheme S1). Overall, this modification increased the yield of the final step from 42 to 63%, which improved the total longest linear yield of BAIT628 from peracetylated mannose to over 5%. Following synthesis, BAIT628 was tested on RAW-Blue cells for its ability to elicit α-man-mediated immunogenicity both from exogenous enzyme as well as TRAMP-C2 cells (Appendix A). The results were consistent with our previous reports of enzyme-directed imidazoquinoline glycoconjugate activity, demonstrating that the newly optimized synthetic route did not alter performance in terms of cell metabolism or enzyme-directed immunogenicity.

### 2.1. Systemically Administered BAIT628 Is Better Tolerated than Imiquimod

With BAIT628 obtained, and retention of α-man-directed activity confirmed, we next examined in vivo safety and tolerability compared with the parent active drug imiquimod (IMQ), whose systemic effects have previously been well characterized [35]. The MTD for BAIT628 was explored using a dosing schedule of daily IP injections. Cohorts included variable doses of BAIT628, an IMQ positive control, or a vehicle negative control administered over 10 days (Figure 2A). As expected for the IMQ-positive control group, serum cytokine levels on day 10 revealed significant (*p* < 0.05), but variable, increases in inflammation, particularly as measured by IL-6 (Figure 2B). This inflammation was also observed as severe dermatitis with coat/skin and limb deterioration in the IMQ group, which required humanely euthanizing multiple IMQ animals prior to the study endpoint. Among the experimental BAIT628 groups, there is a small, but dose-dependent, trend in IL-12 and IL-6 levels. However, this smaller amount of pro-inflammatory cytokine production was not accompanied with the skin or limb abnormalities of the IMQ cohort. Perhaps more importantly, intermediate-dose (0.5 and 1 mg) BAIT628 cohorts also exhibited increased IL-10 levels, which can attenuate the inflammatory response. Overall, the lowest and highest doses of BAIT628 elicited comparably lower serum cytokines. The intermediate doses had modest yet measurable inflammatory cytokine production along with increased anti-inflammatory IL-10. All imidazoquinoline recipients surviving to the study endpoint (including those from the IMQ control) were observed to have normal liver, kidney, and spleen organ weights (Figure 2C). That said, the IMQ cohort exhibited severely degraded organ integrity, particularly for animals meeting humane study endpoint criteria. This included renal and splenic hemorrhage as well as indications of subcapsular hematoma and hemoperitoneum (Appendix A). Overall, production of IL-6 (IMQ) or lack of IL-10 production (BAIT628 5 mg/mL) appeared to correlate with organ pathology. In contrast, BAIT628 at 100 μg and 1 mg/mouse exhibited a unique immunomodulatory response balanced between anti-inflammatory IL-10 and pro-inflammatory cytokines. Further, the 1 mg/mouse dose was well tolerated compared with the same dose of IMQ. Although this study demonstrates that the MTD for systemic BAIT628 is somewhere beyond 5 mg/mouse, we chose 1 mg/mouse for the subsequent tumor growth delay studies. This dose was chosen primarily because it was the highest dose that did not contain any organ abnormalities among any of the mice and would also place the intra-tumoral drug concentration beyond our previously established in vitro EC50. Additionally, this dose produced the maximum amount of IL-10 in the MTD study (the effect was not observed in the 5 mg cohort) and is near the practical limit of what would reasonably scale to a human equivalent dose as a neoadjuvant or in situ vaccination therapy.

### 2.2. BAIT628 Significantly Inhibits Tumor Growth and Exhibits Synergy with α-PD-L1 Checkpoint Inhibitor

With the safety, tolerability, and expanded therapeutic window of BAIT628 established, we next examined BAIT628 efficacy, both as a monotherapy and in synergy with a murinized α-PD-L1 checkpoint inhibitor. For this study, we chose a solid tumor model for prostate cancer. Within the subset of available prostate cancer models, we chose the transgenic adenocarcinoma of the mouse prostate (TRAMP) model [36,37] for its established response to imidazoquinolines including IMQ [38], which is liberated from BAIT628 following metabolism. Importantly, the liberated IMQ is biased towards TLR7 agonism over TLR8. For our model, this detail is non-trivial because in human prostate cancers TLR7 and TLR8 agonism both confer therapeutic benefit [39], whereas in mice, TLR8 is likely anti-inflammatory [40] and can counteract the action of TLR7 agonism [41]. Regardless, this also coincided with our previous work demonstrating that IMQ is effectively trafficked by TRAMP cells [28]. Within the TRAMP series, three stages of androgen independence are available as TRAMP-C1, -C2, and -C3, respectively [29]. Here, we chose the TRAMP-C2 model as this is the most advanced stage of androgen independence that is established to produce palpable tumors when reintroduced to mice [42,43]. TRAMP-C2 prostate tumors exhibit exponential growth and typically reach humane endpoints near 35 days in preclinical models [44]. However, gene expression profiling also indicates that chronic TLR7 stimulation can result in self-tolerance on the order of only 14 days, which would preclude administering drugs evenly throughout the study [45]. To resolve this, we introduced a metronomic dosing strategy that administered drugs every other day with a 5-day space following the 4th dose to achieve a treatment period spanning 3 weeks (Figure 3A).

Mice receiving intra-tumoral BAIT628 monotherapy, or BAIT628 and immune checkpoint inhibitor, exhibited significant tumor growth inhibition compared with the vehicle control group. As expected, tumor growth during and following treatment was uninhibited for the negative control group, resulting in log-phase growth near day 25 and needing to euthanize one animal on day 28 prior to the study endpoint. In contrast, tumor growth was significantly inhibited in the BAIT628 monotherapy and combination therapy cohorts with statistically significant (*p* < 0.05) differences in tumor volumes observed starting on day 25 that continued throughout the duration of the study (Figure 3B). Here, it is also likely that there is synergy between BAIT628 and the α-PD-L1 checkpoint inhibitor; however, the efficacy of the monotherapy obscures this effect in terms of tumor volume. Regardless, all mice in both treatment arms survived to the study endpoint and displayed a therapeutic benefit over control mice.

### 2.3. Intra-Tumoral BAIT628 Is Well Tolerated and Does Not Modulate Serum Cytokines

Because the BAIT mechanism liberates IMQ, we also examined if intra-tumoral administration of BAIT628 produced cytokines similar to systemic administration in the MTD study. To accomplish this, we compared serum cytokines resulting from IT BAIT628 relative to control following administration of the final dose of drug (Figure 3C). In the BAIT628 cohorts, we found what could be modest increases in pro-inflammatory cytokines (TNF and IL-12), although these were not statistically significant. That said, we did observe a modest increase in IL-6, which correlates with polarization of a type I immune response indicative of the parent drug [46]. Furthermore, we again observed the production of IL-10, which could help explain the tolerability of BAIT628. Overall, even though pro-inflammatory cytokine levels were observed in prodrug-treated mice, these were within tolerable ranges and were consistent with body weight measurements demonstrating that the mice have a gradual increase in body weight over the course of the study (Appendix A).

### 2.4. Histological Examination of Tumor Tissues

With tumor volume and systemic serum cytokines characterized, we next characterized the cellular structure of the tumor microenvironment. At a macroscopic level, there were signs of ulceration in the vehicle control group, as is known to occur in mice with untreated TRAMP tumors. H&E staining in the control group also provided the expected morphology of densely packed tumor cells with hyperchromatic nuclei typical of cancerous tissue. Cells were irregularly shaped, characteristic of developed TRAMP tumors, and the lack of large eosinophilic (pink) acellular areas in these tumor sections suggested a dense solid tumor. In contrast, significant differences in the tumor tissue structure of BAIT628-treated cohorts were observed for both monotherapy and combination therapy (Figure 4A). Here, there are several areas of necrotic eosinophilic staining as well as loss of cellular structure and fragmented nuclei. The BAIT628-treated samples are indicative of induced cell death within the tumor cells, with the eosinophilic areas lacking nuclei. That said, BAIT628 was also confirmed to have only a minimal effect on TRAMP cell growth in vitro, indicating that this process is likely mediated by tumor-infiltrating lymphocytes (Appendix A). Regardless, there are also some regions that still show tumor cells with nuclei, indicating incomplete tumor regression. Clusters of smaller cells with dark nuclei, indicative of infiltrating immune cell populations, are also visible. This suggests an immune response initiated by the BAIT628 and/or α-PD-L1, which we subsequently confirmed by flow cytometry.

### 2.5. Immunohistochemical Analysis of Tumor-Infiltrating Immune Cells

One crucial role for immune cells activated by the inflammatory response is their ability for direct cell-mediated killing of cancer cells, and this is no different for solid prostate cancer tumors treated with immunotherapeutics [47]. As such, we investigated the effect of BAIT628 on tumor-infiltrating immune cells, including the frequency of CD4^+^ and effector CD8^+^ T cells as well as CD11c^+^ dendritic cells. In order to determine different immune cell populations, flow cytometry was performed on cell suspensions prepared from snap-frozen tumors resected from each cohort following the tumor growth delay study according to established protocols [48,49]. Analyzing CD4^+^ and CD8+ T cell markers, we found that both subsets were enhanced by BAIT628. These increases were consistent with the diminished tumor volumes observed in the tumor growth delay portion of the study. We also observed a significantly higher frequency of CD11c^+^ cells for both treatment cohorts with a potential additive effect in the combination therapy cohort (Figure 4B). Regardless, these results were again consistent with treating prostate cancer tumors with TLR7 agonists.

## 3. Methods

### 3.1. Antibodies and Drugs

See Appendix A for a table of biologics and BAIT628 synthesis. PE anti-mouse CD4, PE Rat IgG2b κ Isotype Ctrl Antibody, Brilliant Violet 421™ Rat IgG2a κ Isotype Ctrl Antibody, and APC Rat IgG2a κ Isotype Ctrl Antibody were purchased from Biolegend. Bioscience brand Monoclonal anti-CD8a Alexa Fluor 488, Rat IgG2a kappa Isotype Control Alexa Fluor 488, Rabbit IgG Isotype Control, and Mouse IgG2a-κ Isotype Control Alexa Fluor™ 488 were purchased from Thermo Fisher. Hamster Anti-Mouse CD11c APC-Cy7 was purchased from BD bioscience (Appendix A). Anti-mPD-L1-mIgG1e3 (10F.9G2) was purchased from Invivogen, San Diego, CA, USA. The improved BAIT628 synthesis was developed in-house with an additional compound synthesized by PharmaInventor (Toronto, ON, Canada).

### 3.2. Resazurin Assay

The effect of BAIT628 on cell viability was assessed using a resazurin reduction assay to observe reduction of blue non-fluorescent resazurin reagent to pink resorufin by intracellular oxidoreductases [50]. TRAMP-C2 cells were seeded (2 × 10^3^ cells/well) and allowed to adhere for 24 h. Cells were then incubated with BAIT628 (1 nM–100 μM) for an additional 24 h alongside positive (100 μM imiquimod) or negative (vehicle) controls. Resazurin solution (10 μL) was added and incubated for 6 h at 37 °C and 5% CO_2_. Absorbance (570 and 600 nm) was measured. Control wells were standardized as 100% viability and compared with other treated wells. Cell viability percentage was determined as: (absorbance of treated cells)/(absorbance of matched vehicle controls) × 100 (Appendix A).

### 3.3. RAW-Blue Cell Culture

RAW-Blue cells (Appendix A) were purchased from Invivogen. Cells were maintained in high-glucose (4.5 g/L) Dulbecco’s modified Eagle medium (DMEM) with 4 mM L-glutamine, 1 mM sodium pyruvate, 10% heat-inactivated fetal bovine serum (Corning), along with 100 μg/mL Normocin, 100 μg/mL zeocin (Invivogen) selective antibiotic, 100 U/mL penicillin, and 100 μg/mL streptomycin (Sigma-Aldrich, St. Louis, MO, USA), in a 5% CO_2_ atmosphere at 37 °C. Growth media were changed every 3–4 days. Cells were passaged at 80% confluence and used per manufacturer instructions.

### 3.4. TRAMP-C2 Cell Culture

The TRAMP-C2 cell line (Appendix A) was purchased from American Type Culture Collection (ATCC). TRAMP-C2 cells were maintained in T-75 culture flasks (VWR) using high-glucose (4.5 g/L) DMEM supplemented with 4 mM L-glutamine, 1 mM sodium pyruvate, 0.005 mg/mL recombinant human insulin (Thermo Fisher), 10 nM dehydroisoandrosterone (VWR, Radnor, PA, USA), 5% heat-inactivated fetal bovine serum, 5% Nu Serum IV (Corning, NY, USA), 100 U/mL penicillin, and 100 μg/mL streptomycin (Sigma-Aldrich). Cells were incubated in 5% CO_2_ atmosphere at 37 °C with growth media changed every 3–4 days. Cells were passaged at 80% confluence using trypsin solution (Sigma-Aldrich) per manufacturer instructions. Trypsinized cells were washed with phosphate-buffered saline (PBS), and seeded at a density of 1.0 × 10^6^ cells in 20 mL of complete media in a T-75 culture flask (VWR).

### 3.5. Cancer-Mediated Immunogenicity of BAIT628

BAIT628 was added to TRAMP-C2 cells seeded in a 96-well plate and incubated for 24 h. Control experiments included wells that did not contain cells (negative control) as well as wells that contained BAIT628 with exogenous 0.1 U/mL α-mannosidase enzyme (positive control). The resulting immunogenicity was measured by the addition of 20 μL of cancer cell supernatants, exogenous α-mannosidase samples, or BAIT628 solution alone to RAW-Blue cells. Cells were incubated for 24 h before assaying immunogenicity using Quanti-Blue reagent (1.168 mg/mL 5-bromo-4-chloro-3’-indolylphosphate *p*-toluidine salt in 1 M aqueous diethanolamine) to assay immunogenicity as secreted alkaline phosphatase production (measured as absorbance at 620 nm) per manufacturer instructions (Invivogen). Activity was normalized relative to the direct addition of imiquimod (positive control) or PBS (negative control) to RAW-Blue cells (Appendix A).

### 3.6. Animals

Animals were maintained at Miami University, and all procedures were performed under the approved IACUC protocol 1077_2025_Aug. Male C57BL/6 (Black 6) mice were purchased from Charles River. Mice were equilibrated and aged between 8 and 12 weeks before beginning experiments on day 0. All animals had free access to chow, water, and environmental enrichment. Animal behavior and clinical body scores were assessed daily. Body weight was recorded every other day. Drugs were administered, blood samples were taken (via saphenous vein), and tumor volumes were measured at timepoints indicated in each subsequent experiment. Mice were humanely euthanized upon reaching study or established humane endpoints similar to those reviewed here [51]. Data from all study animals were included in the final analyses, and both the MTD and tumor growth delay studies followed ARRIVE guidelines [52].

### 3.7. Safety and Tolerability of Systemic BAIT628

The safety and tolerability of systemically administered BAIT628 was examined in a maximum tolerable dose (MTD) study. On day 0, and throughout the study, male C57BL/6 mice were assessed via a health monitoring sheet, including body scoring, body weight, and overall health condition. A total of 36 mice were divided into 6 cohorts (n = 6) such that variability in starting body weight between cohorts was minimized. Study arms were randomly assigned as Cohort 1: vehicle (negative control), Cohorts 2–5: BAIT628 at 0.1, 0.5, 1, and 5 mg/mouse, and Cohort 6: imiquimod at 1 mg/mouse (positive control). All drugs were administered intraperitoneally (IP) daily for 10 days (Figure 2A). Blood was collected via the saphenous vein 6 h following the final drug dose. The duration of the study spanned 30 days or until reaching animal welfare study endpoints. Cytokine levels were quantified from sera samples using a cytometric bead array kit (Figure 2B). Body condition was monitored, and skin conditions were graded and photographed (Appendix A). Organs were removed, weighed (Figure 2C), and photographed (Appendix A).

### 3.8. TRAMP-C2 Tumor Growth and Treatment

On day 0 (Figure 3A), mice were implanted with prostate cancer cells in their shaven right hind flank by subcutaneous injection of 2.5 × 10^6^ TRAMP-C2 cells in 200 μL of a 1:1 mixture of PBS and Matrigel (Corning). Tumor volumes were measured with calipers and calculated as volume = 0.5 × D × d^2^, where D is the larger diameter of the tumor and d is the smaller diameter. When palpable tumors reached between 200 and 300 mm^3^, 9 tumor-bearing mice were categorized in 3 arms such that the starting average tumor volume was consistent between groups. Groups were then randomly assigned as Group 1: Negative vehicle control, Group 2: BAIT628 (1 mg/mouse), and Group 3: BAIT628 (1 mg/mouse) with anti-mPD-L1-mIgG1e3 (10F.9G2) checkpoint inhibitor (50 μg/mouse). Drugs were administered via 50 μL intra-tumoral injection 7 times over 3 weeks (every other day with a 5-day space after the 4th dose to hedge against tolerance). Tumor volume (Figure 3B) and body weight (Appendix A) were assessed twice per week. Following the last day of drug administration, blood was taken, and serum cytokines were analyzed by a cytometric bead array (Figure 3C). Mouse organs and tumors were harvested and characterized via histological and immunohistochemistry assessment. Mice meeting humane endpoint criteria or that did not produce tumors were humanely euthanized.

### 3.9. Cytometric Bead Array of Serum Cytokines

Mouse serum samples were prepared following blood draw from the last dose of drug administration at the days indicated in the MTD and tumor growth delay experiments. Following collection, blood was centrifuged (15 min, 1500× *g*, 4 °C), and sera samples were stored in cryovials at −80 °C. On the day of analysis, samples were thawed, and pro-inflammatory and anti-inflammatory cytokine levels were determined using a cytometric bead array mouse inflammatory kit according to the manufacturer’s instructions (BD Biosciences, Franklin Lakes, NJ, USA). In brief, the mouse inflammation capture beads for each cytokine were mixed before using them in the assay. Standard curves were generated for each cytokine from 20 to 5000 pg/mL and resulted in limits of detection of 117, 180, 125, 188, 355, and 1100 pg/mL for IL-6, IL-10, MCP-1, IFN-γ, TNF-α, and IL12p70, respectively. To perform the assay, 50 μL of mixed capture beads was added to 50 μL of diluted cytokine standard solution, controls, or unknown samples followed by 50 μL of mouse inflammation PE detection reagent. Samples were incubated for two hours in a dark at room temperature before washing with 1 mL of wash buffer and resuspending in 300 μL of wash buffer for characterization by flow cytometry (FC) [53].

### 3.10. Brightfield Tumor Histology

Histopathological analysis was performed by fixing tumor tissue in fixing and cryoprotective solution (4% formalin and 30% sucrose in PBS at 4 °C for 1 h) [54]. The samples were then embedded in optimal cutting temperature (O.C.T.) compound (Thermo Fisher, Waltham, MA, USA), frozen in a dry ice–acetone mixture, and stored at −80 °C until sectioning. Just prior to cryostat sectioning, frozen tissue blocks were equilibrated to −20 °C. Tissue sections of 5–10 µm thickness were collected onto immunohistochemistry-grade positively charged glass slides and stained by immersion in hematoxylin solution (Abcam, Cambridge, UK) for 1–2 min, followed by rinsing in distilled water for at least 10 min in several water changes. Adequate bluing reagent was applied to cover the tissue section completely and incubated for 20 s followed by rinsing in two changes of distilled water. The tissue sections were then dehydrated using four changes of alcohol (95%, 95%, 100%, and 100%) for 2 min each followed by covering tissue sections in eosin Y solution (modified alcoholic) and incubating for 2 min. Finally, the tissue sections were dehydrated in four changes of absolute alcohol (95%, 95%, 100%, and 100%), cleared in three changes of xylene, and mounted with mounting medium (VECTASHIELD Antifade Mounting Media). The resulting samples were imaged via brightfield microscopy using a 20× air objective (Figure 4A).

### 3.11. Tumor Homogenization for Flow Cytometry

Harvested prostate tumors were minced with scissors and placed in a Petri dish containing 5 mL of ice-cold DPBS supplemented with 0.5% BSA. Next, single-cell suspensions were prepared by incubating with a prewarmed dissociation buffer (collagenase IV 2 mg/mL, DNase I 1000 4 U/mL in RPMI). Minced tumor and buffer were pipetted into 15 mL centrifuge tubes, and an additional 10 mL of dissociation buffer was added. Samples were incubated at 37 °C and shaken every 10 min for a total of 30 min. The resulting cell suspension was centrifuged (5 min at 300× *g*), and the supernatant was discarded. The cell pellets were resuspended in 10 mL of DPBS supplemented with 0.5% BSA and filtered through a 70 μm cell strainer into a 50 mL centrifuge tube. Cells density was assessed by counting with trypan blue and a hemocytometer before blocking and staining [55].

### 3.12. Tumor Homogenate Immunohistochemistry

Tumor cell suspensions were kept on ice during the whole process with 1 × 10^6^ cells being used for each immune marker experiment. To reduce non-specific binding, 10 µL of Fc block CD16/CD32 (1:50 dilution) was added to the cells resuspended in 100 µL of ice-cold FACS buffer and incubated for one hour. The cell suspensions were centrifuged, and the supernatant was discarded. The cell pellets were resuspended in 100 µL of FACS buffer, and immune-staining antibodies solution was added to each sample at the working dilutions noted above before incubation on ice for one hour. Cells were then centrifuged (5 min at 300× *g*), the cell supernatant was discarded, and then the pellet was resuspended in 100 µL of FACS buffer for further FC analysis. Cells were first gated based on forward scatter (FSC) and side scatter (SSC), followed by further refinement using FSC area (FSC-A) versus FSC height (FSC-H) to discern singlet cells. Relevant fluorescent channels were examined compared with isotype controls to determine cell populations that expressed each examined cell marker (CD4, CD8, CD11c) [56]. Gating was determined relative to isotype controls to determine the percentage of cells expressing each marker (Figure 4B).

### 3.13. Statistical Analysis

Statistical analysis was performed in Microsoft Excel. A two-tailed t test assuming unequal variances was used to compare the vehicle group and the treatment groups. All experiments were performed at least three times, and data are presented as the mean ± SEM.

## 4. Conclusions

In summary, we have demonstrated the in vivo safety, tolerability, and efficacy of BAIT628 as an anti-cancer and enzyme-directed immunostimulant in a TRAMP-C2 prostate cancer model system. Our MTD study suggests that one reason for BATI628’s profound tolerability (5 mg/mouse/day, IP) over the parent IMQ immunostimulant could be the production of anti-inflammatory cytokines (such as IL-10), which are associated with the mannose sugar used in the prodrug cage. Histological analysis of tumor sections through H&E staining following the tumor growth delay study indicates that BAIT628 is effective in reducing tumor burden and likely acts in synergy with the α-PD-L1 checkpoint blockade to reduce the nuclear density of cancer cells within solid tumors. Quantifying immune cell subsets by flow cytometry reveals that tumor-infiltrating immune cell populations are enhanced by BAIT628 both as mono- and combination therapies. While we observed increases in CD4^+^, CD8^+^, and CD11c^+^ cells, only CD11c^+^ cells had a synergistic increase for the combination therapy. While these results are promising, it is also important to consider some key limitations of this first-in-mouse, proof-of-concept study. First, while this study does establish efficacy of BAIT628, it only performs this at a single, relatively high dose and at a small sample size (n = 6 and n = 3). This could obfuscate synergistic effects with combination therapies like checkpoint inhibitors and precluded a more detailed characterization of the systemic immune response such as at the cellular organ level (e.g., splenocyte populations). Generally, a larger, more comprehensive study would be needed to conclusively determine if the effect of the mannoside caging group on imidazoquinolinone glycoconjugate PK/PD is similar to its previously established effect on chemotherapeutic drugs. Regardless, because BAIT628 is efficacious in vivo, and well tolerated systemically, we envision this technique could be applied to other cancer models requiring more challenging routes of drug delivery or other imidazoquinoline core structures that have previously been disregarded due to their immune-related adverse events. In the future, we plan to test BAIT628, alongside other imidazoquinolines optimized for cancer cell metabolism [57], in some of the many multidrug-resistant cancers containing metabolic irregularities that would further enhance the BAIT mechanism of action.

## Figures and Tables

**Figure 1 pharmaceuticals-18-00804-f001:**
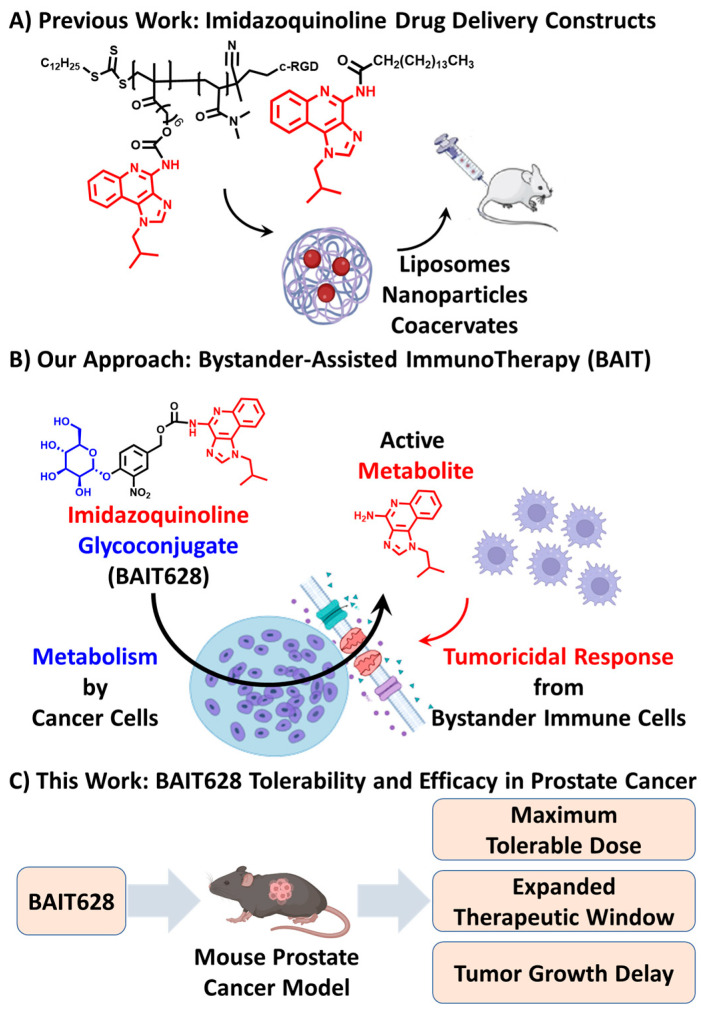
Development of the drug candidate BAIT628 used in this work. (**A**) Many groups have formulated imidazoquinolines as macromolecular nanoparticles, liposomes, hydrogels, coacervates, or other drug delivery constructs. (**B**) Our group demonstrated a small-molecule approach to imidazoquinoline drug delivery using a glycoconjugate prodrug (BAIT628). The chemical structure contains a TLR 7/8 agonist imidazoquinoline (**red**) with activity modulated by the attachment of a mannose substituent (**blue**) through a self-immolative spacer (**black**). Both the mannose carbohydrate and linker are removed by the action of intra-cellular α-mannosidases, thus liberating active imidazoquinoline. We have also determined that the active imidazoquinoline is effluxed to the extracellular space via MDR1 and can activate immune cells in local proximity in a net process termed Bystander-Assisted ImmunoTherapy (BAIT). (**C**) This work is the first-in-mouse study of a BAIT prodrug where we evaluate safety and tolerability, as well as tumor growth delay efficacy in a mouse TRAMP prostate cancer model.

**Figure 2 pharmaceuticals-18-00804-f002:**
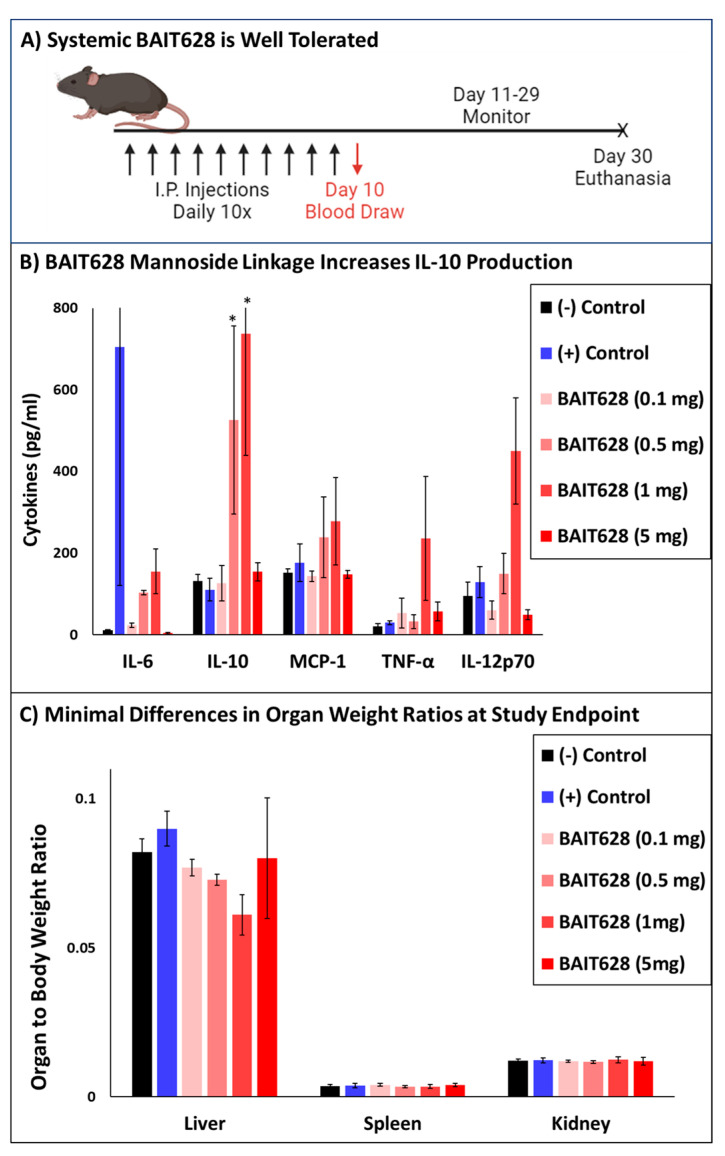
Variable doses of BAIT628 compared with a 1 mg/mouse imiquimod (IMQ) positive control. (**A**) Drugs were administered IP at the indicated doses daily for 10 days. Blood draw on day 10 was followed by an analysis of (**B**) cytokine levels in serum. Data are presented as the mean ± SEM (pg/mL) for sera collected from 6 individual mice per cohort. * indicates significant differences (*p* < 0.05) for IL-10 compared with the negative control group (n = 6). Following drug administration, mice were monitored for health, body condition, and body weight over 20 days before. (**C**) Mouse organ (liver, spleen, kidney) weight ratio relative to body weight. Mice received BAIT628 at various doses (0.1, 0.5, 1.0, 5mg) and results compared to vehicle group and positive control (1 mg/mouse imiquimod). Data are presented as mean ± SEM. See Appendix A for organ images.

**Figure 3 pharmaceuticals-18-00804-f003:**
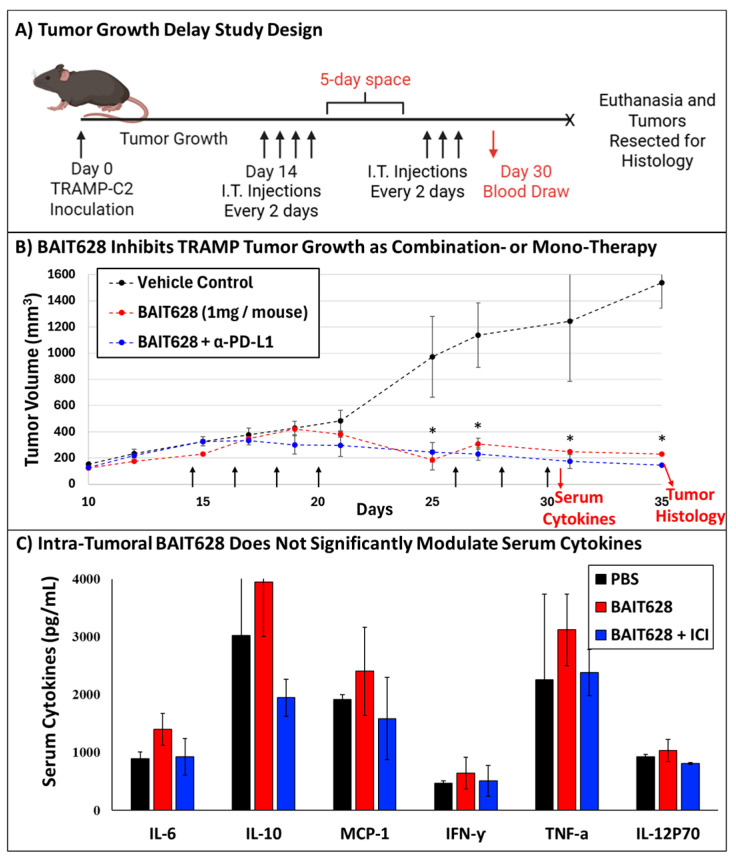
TRAMP-C2 tumor growth delay elicited by BAIT628 or BAIT628 combination therapy with murinized α-PD-L1 checkpoint inhibitor. (**A**) Mice with palpable tumors were administered drugs via intra-tumoral injection every other day over 3 weeks. Treatment regimens started on day 14, and a 5-day space was included after the 4th dose. Overall health and weight were monitored alongside (**B**) changes in tumor volume. Here, groups receiving BAIT628 (**red**) or BAIT628 in combination with checkpoint inhibitor (**blue**) had reduced tumor volumes relative to vehicle control-treated mice (**black**). Beginning on day 25, and continuing through the study endpoint, this change in volume was significant (n = 3, * *p* < 0.05). Results are represented as the mean ± SEM. (**C**) Serum cytokine analysis revealed potentially elevated IL-10 for BAIT628 similar to the MTD study, although cytokine changes were not significant compared with the control.

**Figure 4 pharmaceuticals-18-00804-f004:**
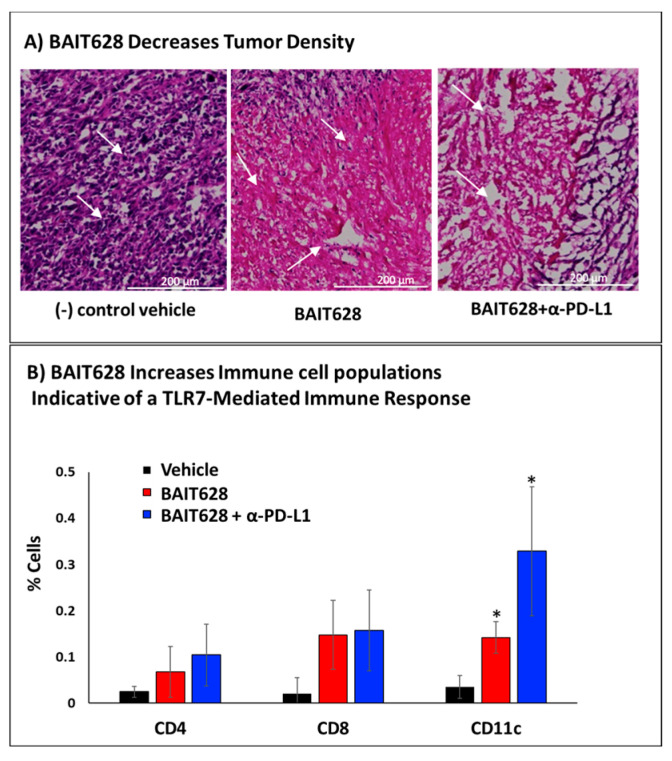
Characterization of the BAIT-treated tumor microenvironment. (**A**) Histology of tumor samples in mice treated with vehicle control, BAIT628 monotherapy, or BAIT628 and murinized α-PD-L1 combination therapy. At the conclusion of the tumor growth delay study, tumors were excised immediately following euthanasia and flash frozen. Following cryo-sectioning and H&E staining (see experimental). White arrows indicate areas in tumor sections where tumor cells are fragmented and have lost their nuclei density or structural integrity of the solid tumor compared with the control. Images are representative of each cohort (n = 3) and were obtained using a brightfield polarized microscope with a 20x air objective lens. (**B**) Intra-tumoral immune cell populations elicited by BAIT628 (**red**) or BAIT628 immune checkpoint inhibitor combination therapy (**blue**) cohorts relative to vehicle control (**black**) across CD11c^+^ dendritic cells, CD4^+^, or CD8^+^ T cells. Comparing these cell populations revealed a synergistic enhancement of CD11c^+^ cells upon the addition of checkpoint inhibitor but a modest decrease in CD4^+^ cells. Changes in CD8^+^ cells were increased relative to the control but not significantly different between the treatment cohorts. * *p* < 0.05 and error bars are the standard deviations of the mean for experiments repeated in biological triplicate.

## Data Availability

Data sets from each study are available upon email request to the corresponding author (mancinr@miamioh.edu).

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
