# Peer review of "Performance of Imidazoquinoline Glycoconjugate BAIT628 as a TLR7 Agonist Prodrug for Prostate Cancer"

_pharmaceuticals, 2025, doi:10.3390/ph18060804_

Round 1
Reviewer 1 Report
Comments and Suggestions for Authors
- The clear abstract outlines the aim, methodology, and key findings.
- It is suggested that the authors briefly clarify the differences between TLR7 and TLR8 selectivity and relevance to prostate cancer.
- Authors are advised to cite this recent reference https://doi.org/10.1016/j.bpc.2025.107435 in the second paragraph of the introduction.
- The authors should justify why the 1 mg/mouse BAIT628 dose was chosen and give more detailed results and explanations about the dose.
- It is suggested that the authors provide a dose-response efficacy study or IC50 value.
- Why authors have chosen a relatively small sample size (n = 3) for the tumor efficacy study should be justified, or its limitations mentioned in the discussion.
- The manuscript refers to IL-10 contributing to tolerability, but a direct correlation analysis between IL-10 levels and clinical score or organ toxicity is not provided. Authors are suggested for this portion, as this would strengthen the conclusion.
- The cytokine assay method (cytometric bead array) should specify detection limits and sensitivity ranges for each cytokine.
- Authors are advised to add a scale bar for histology images (Figure 4A).
Reviewer 2 Report
Comments and Suggestions for Authors
This manuscript presents an in vivo evaluation of BAIT628, an imidazoquinoline-based glycoconjugate prodrug, for prostate cancer therapy.
Below are some comments:
- Citations [14], [15], and [34] — all of which are prior works from Mancini RJ’s group. [14] is the foundational work, [15] [34] are the follow-up works. This paper relies heavily on the authors’ prior work to establish the novelty and mechanism of BAIT628.
- Line 68, the authors previously demonstrated the BAIT mechanism in multiple cancer types in vitro, including melanoma, breast, and prostate cancer. It would be appreciated if the author could include the reasons for proceeding in vivo, specifically with prostate cancer.
- Line 249, although IL-10 is discussed as a tolerability factor, the study does not analyze CD206 activation in tumor-associated macrophages, despite citing relevant literature.
- Figure 4B, The gating strategy is mentioned but not shown. Adding supplementary figures would improve reproducibility.
- Figure 4B, The statistical comparison (p value) between the BAIT628 mono and combination groups is not shown in the figure.
- Line 401, 402, the statement "a significantly higher frequency of" is not sufficiently supported by Figure 4b.
- The manuscript claims a synergistic or additive effect for the combination of BAIT628 and α-PD-L1 checkpoint inhibition; however, this is not convincingly supported by the presented tumor volume, cytokine, or immune infiltration data. No statistical comparison is made between the mono and combination, and the trends in CD8+ and CD4+ T cells are largely similar. It would be better to have a further explanation.
Reviewer 3 Report
Comments and Suggestions for Authors
Additional experiments are needed to characterize the BAIT628 biodistribution, pharmacokinetics and organs toxicity profile
1) Biodistribution: To evaluate the safety and dosing justification, even if the MTD and efficacy studies look good, it's highly important to show data how the compound distributes across tissues. In addition, since BAIT628 acts through immune modulation and tumor targeting, showing that it localizes to the tumor and relevant immune organs (e.g., spleen, lymph nodes) strengthens your mechanistic claims. If BAIT628 is better tolerated than IMQ, showing how its distribution differs (e.g., reduced systemic exposure, slower release) would quantitatively support your hypothesis about the mannose cage or delivery method
2) Pharmacokinetics: will add an essential data for how long it remains active in circulation.
3) Immunohistochemistry analysis is essential to analyze the toxicity on the organs tissue level with all immune markers (CD4+, CD8+, and CD11c+ cells, only CD11c+) for immune cells infiltration and not just tumor. Since just evaluating organs weight is not enough to evaluate the tissue toxicity.
Round 2
Reviewer 3 Report
Comments and Suggestions for Authors
While I understand that biodistribution, pharmacokinetics, and detailed immunohistochemistry are beyond the scope of this proof-of-concept study, I recommend clearly stating these as limitations in the discussion. This will ensure transparency and help contextualize the findings when considering future translational work. In other words:
-
PK of the prodrug vs parent drug –
While Cavanagh 2017 covers the parent IMQ, a prodrug (especially with a sugar-based cage) can significantly alter bioavailability, activation kinetics, and immune engagement. Authors should at least acknowledge this limitation in the discussion, even if data isn’t available. -
Reliance on tumor growth + organ weight without immunohistochemistry:
Evaluating immune cell infiltration only at the tumor level, and systemic toxicity only by organ weight leaves a real knowledge gap. While full IHC in all organs may be beyond scope, an acknowledgment in limitations would be expected in a strong manuscript.
Author Response
In the conclusions section of our revised manuscript we now include statements on the limitations of the study (PK of prodrug vs. parent and organ immunohistochemistry) pointed out by the reviewer.